# Relationship Between Sleep and Immunology in Attention Deficit Hyperactivity Disorder

**DOI:** 10.3390/ijms26167967

**Published:** 2025-08-18

**Authors:** Julia Jaromirska, Marcin Sochal, Dominik Strzelecki, Piotr Białasiewicz, Agata Gabryelska

**Affiliations:** 1Department of Sleep Medicine and Metabolic Disorders, Medical University of Lodz, 92-251 Lodz, Poland; julia.jaromirska@stud.umed.lodz.pl (J.J.); marcin.sochal@umed.lodz.pl (M.S.); piotr.bialasiewicz@umed.lodz.pl (P.B.); 2Department of Affective and Psychotic Disorders, Medical University of Lodz, 92-251 Lodz, Poland; dominik.strzelecki@umed.lodz.pl

**Keywords:** sleep, immunology, attention deficit hyperactivity disorder, neurodevelopment

## Abstract

Attention deficit hyperactivity disorder (ADHD) is a complex neurodevelopmental disorder that not only affects attention and behavior but is also intricately linked with sleep disturbances and immune system dysregulation. Recent research highlights that individuals with ADHD frequently experience sleep problems, which in turn exacerbate ADHD symptoms and contribute to cognitive and emotional difficulties. Immunological alterations, including elevated proinflammatory cytokines and hypothalamic–pituitary–adrenal axis dysfunction, have been observed among ADHD patients, suggesting a biological interplay between inflammation, sleep, and neurodevelopment. Genetic and environmental factors further modulate these relationships, influencing the onset and progression of the disorder. Thus, there is a need to find a key connecting such topics and the most vulnerable subjects in order to contribute towards a more personalized approach. This review examines the complex relationships between sleep, immunology, and ADHD, and explores the underlying mechanisms that involve circadian rhythm genes, neuroinflammation, and neurotransmitter imbalances. Our review outlines therapeutic strategies, emphasizing the importance of integrated pharmacological, behavioral, and lifestyle interventions to improve sleep quality, regulate immune responses, and ultimately enhance the overall management of ADHD.

## 1. Introduction

Attention deficit hyperactivity disorder (ADHD) is a neurodevelopmental pediatric-onset disorder with a prevalence of 5.29% worldwide. ADHD symptoms start to occur in childhood; however, some patients are diagnosed as adults, when the compensation mechanisms are no longer capable of allowing individuals to cope with the symptoms. Additionally, awareness about the disorder has significantly increased in recent decades, not only among doctors, but also among the general population, which has contributed to a greater number of ADHD diagnoses in the adult population. Although research on the topic indicates significant variability in the prevalence of ADHD, this phenomenon can be explained by methodological discrepancies and underdiagnosis in some populations. It is unlikely that the number of incidences has increased over the decades [1]. According to the *Diagnostic and Statistical Manual of Mental Disorders*, 5th Edition, Text Revision, ADHD is a persistent pattern of inattention and/or hyperactivity-impulsivity that interferes with functioning or development; the main diagnostic criteria include inattention, hyperactivity, and impulsivity. The main types of the disorder are, as follows: predominantly inattentive; predominantly impulsive/hyperactive; and combined presentation. In general, possibly due to the influence of upbringing, there are gender differences in the clinical presentation—girls with ADHD were reported to exhibit more significant intellectual difficulties, exhibited less hyperactivity, and engaged in fewer externalizing behaviors [2]. Patients with ADHD are more prone to experiencing difficulties with: quality of life (measured by lower life productivity, psychological health, and life outlook) [3]; relationships with peers (measured by lower level of the perceived quality of social relations in comparison to healthy peers through sociometric status) [4] and family (depicted by lower Family Crisis Oriented Personal Evaluation Scale scores, thus greater family dysfunction) [5]; and school performance (including grade retention OR = 3.58, secondary school graduation failure OR = 2.41, or lower level diploma OR = 3.0) [6,7]. The disorder is marked by increased comorbidity with learning disorders and autism spectrum [8]; autism spectrum and tic disorders [9]; tic disorders and obsessive–compulsive disorder [10]; obsessive–compulsive disorder and mood disorders [11]; and mood disorders [12]. The current treatment for ADHD disorder comprises both pharmacological and therapeutical approaches. Efficacious drugs used in the disorder include psychostimulants (methylphenidate, dexamphetamine, mixed amphetamine salts, and lisdexamfetamine) and non-psychostimulants (atomoxetine, viloxazine, bupropion). In contrast, non-pharmacological treatment consists of cognitive–behavioral therapy, neurofeedback training, and cognitive training interventions [13]. Properly treated patients usually have alleviated symptoms of ADHD; however, treatment-related adverse side effects are common and inconvenient [14]. The average ADHD treatment induces approximately six side effects [15].

Patients with ADHD constitute a heterogeneous group of people with different demands and needs with respect to improving their lives. One of the most prevalent problems (reported by 25–50% of patients with ADHD) includes sleep disturbances, and, thus, decreased quality of life, further impairment of cognitive performance, and more significant attention deficits [16]. In recent years, researchers have investigated the connection between these symptoms and the immunological aspects of the ADHD profile; both share the same abnormalities in immunity and their influence on each other can be largely dependent on immunological processes. Thus, this review aims to describe current knowledge about the associations between sleep, ADHD and their associated immune pathways, and to highlight the possible directions for further research in this area.

## 2. Immunological Aspects of ADHD

### 2.1. General Overview of Immune Changes Among ADHD Subjects

In recent years, more research has explored the association between ADHD and aspects of immunology. In general, people with ADHD have altered immunological profiles, including typically higher levels of interleukin 6 (IL-6, known to influence microglia), IL-1-beta (a key proinflammatory cytokine known to influence microglia) [14], IL-1-beta (a key proinflammatory cytokine), and tumor necrosis factor (TNF) alpha (related to a weak immune response), which negatively correlate with symptom severity [17]. This means that ADHD subjects are more prone to developing immune disorders due to cytokine disruption. MicroRNA-200b-3p or taurine intake seems to lower those cytokines [18], which may be a promising pathway in the search for a treatment target. As ADHD is perceived as a neurodevelopmental disorder without a strictly defined genesis, the evaluation of particular microRNAs might be helpful in unraveling those biochemical relations. At a molecular level, ADHD contributes to the increase in myeloid dendritic cells, monocytes, and granulocytes. In the saliva of ADHD subjects, there were significantly elevated levels of alpha-amylase, immunoglobulin A, and immunoglobulin M, although no differences in morning cortisol were detected [19]. However, according to a systematic review by Chang et al., ADHD individuals have lower cumulative cortisol with lower morning levels [20]. This means that stress markers seem to be increased among ADHD subjects, pointing to a plausible role of such an enhanced immune response. Children with ADHD were found to have a downregulated hypothalamic–pituitary–adrenal (HPA) axis, as shown in the presence of single nucleotide polymorphisms (especially rs9470080) in FK506 binding protein 5—involved in immunoregulation, e.g., via interactions with corticoid receptor complexes—and lower diurnal cortisol levels in comparison to healthy children [21]. In addition, glucocorticoid receptors TthIIII_rs10052957 [22], NR3C1-I_rs10482605, and ER22/23EK_rs6189/rs6190 variants were associated with ADHD diagnosis. These signs of subclinical inflammation in morphology, cytokine levels, or hormone disbalances might be connected with the etiology of ADHD, which so far has not been explained [23].

Both genetic and environmental factors contribute to shaping this profile. In terms of genetics, the offspring of ADHD parents are more prone to developing psychiatric disorders, including ADHD—up to 88% in terms of the formal heritability [24]. In terms of environment, many risk factors have been suggested and confirmed to be associated with ADHD onset, including: maternal pre-pregnancy obesity, childhood eczema, hypertensive disorders during pregnancy and pre-eclampsia, maternal acetaminophen exposure during pregnancy, maternal smoking during pregnancy, childhood asthma, maternal pre-pregnancy overweight, and serum vitamin D [25]. The influence of the mother’s immunology was shown to be connected, especially in the context of the disorder’s predisposition, as well as the offspring’s immunology, as a triggering factor for ADHD occurrence. Environmental factors, such as increased air pollution, suggest that economically disadvantaged populations might be predisposed to the syndrome [26]. Although many potential connections are not well-established, and the human immune system remains incompletely understood, some preliminary conclusions can be drawn [27].

### 2.2. Impaired Immunity Before Birth and ADHD Symptoms

Differences can occur in the prenatal development of the immune system between subjects who will later develop neurodevelopmental disorders and those who will not. Although the heritability of ADHD is high, there are no confirmed genes, including proinflammatory genes, responsible for this. The most promising by far are those connected with dopamine and serotonin circuits. One polymorphism of dopamine transporter was proved to be associated with an almost three times greater prevalence of ADHD compared to children without this allele. The presence of dopamine receptor D4 polymorphism was shown to contribute to the increased risk of ADHD development. Both polymorphisms can be inherited but may also be influenced by prenatal smoking exposure. The serotonin-transporter-linked polymorphic region (5-HTTLPR) [28] was found to be a bridge between maternal anxiety and negative emotionality, enhancing the vulnerability of the fetus to environmental risk factors for ADHD. Through the same polymorphism, maternal smoking might affect the fetal serotonin system, causing the fetus’ vulnerability to emotional problems [29,30,31]. Insights into the genetic risk of developing ADHD can be enhanced by an evaluation of pregnancy history. Both factors can cause innate disruption. Mothers with diverse inflammatory factors (such as obesity, autoimmune disease, infection during pregnancy, prolonged stress [32], smoking, and sleep disorders [33]) are more likely to give birth to a child with subsequent ADHD. This process probably occurs, among other processes, through the activation of toll-like receptors [34], similar to other autoimmune disorders [35]. In prenatal offspring, it induces neuroinflammation, affecting brain development and synaptic transmission [36], which is consistent with the neuroinflammation hypothesis in ADHD pathophysiology, suggesting that inflammation of nervous tissue can have a detrimental effect on the development of psychiatric disorders. As the nervous system starts its growth in the 3–4th week after conception, any unfavorable factor affecting neurogenesis and subsequent formations from this time have an impact on post-natal life, e.g., in forming significantly smaller total cerebral (lower by 3.2%) and cerebellar (lower by 3.5%) brain volume in ADHD children, with more detrimental changes among unmedicated ones (cerebral volume lower by 5.8%, cerebellar volume lower by 6.2%) based on magnetic resonance images. It is notable that similar changes occur in children and adults with ADHD; the extent of the morphologic changes correlate with specific subtypes of the disorder in ADHD children with more detrimental changes among unmedicated ones (cerebral volume lower by 5.8%, cerebellar volume lower by 6.2%) based on magnetic resonance images [37]. In an animal model, the application of anti-inflammatory treatment to the mother ameliorated abnormal locomotor activity, indicating the protection of neurodevelopment and creating an opportunity to lessen the risk of ADHD occurrence in situations of excess maternal stress [38,39]. 

### 2.3. Impaired Immunity After Birth and ADHD Symptoms

After labor, immunology-affecting factors can also predispose one to develop ADHD; the best-described among these is the gut microbiome influence. A longitudinal study of over 16,000 Swedish children revealed that infant exposure to infectious agents and antibiotic usage, which affect the human microbiome, is related to a higher risk of ADHD. Gut microbiome imbalance in later life among people with ADHD was reported to be amended. However, in older patients, the preference for a non-healthy diet can be the reason for such a connection. It is notable that a dietary intervention in the form of vitamin D and vitamin D with magnesium as co-adjuvant treatment improved ADHD pharmacological treatment [40]. Influence on the gut microbiome is also exerted by psychostimulant medications, pointing to differences in taxonomic beta diversity and lower evenness [41,42]. In addition, coexisting diseases derived from the dysregulated immune system are more commonly present among ADHD patients than the rest of the population, with the most common including asthma [43,44] and autoimmunological diseases such as allergy [45,46], coeliac disease [47,48], diabetes mellitus type 1 [49,50], and thyroid abnormalities [51]. Although many of the precise connections are still unknown, it can be postulated that these disorders work together to create a self-perpetuating cycle that leads to a dysregulated immune system.

### 2.4. Conclusions of the Section

As described above, subjects with ADHD have an altered immunological system because of both genetic and environmental factors. Therefore, the strongest scientific risk factors should be taken into consideration while assessing the patient. It is important to note that the majority of studies did not stipulate reliable information on which ADHD phenotypes the factors affect the most. As the awareness of risk factors is still not widespread, preventive knowledge and actions should be expanded upon. A summary of this section is shown in Table 1.

## 3. Importance of Sleep and Sleep Disruption in ADHD

Sleep is one of the most critical factors determining the quality of life, as its normal duration is around 1/3 of the day. Sleep is a recovery state of the sleep–wake cycle; thus, the nervous system’s development largely relies on it. Thanks to the circadian rhythm, a 24 h oscillation occurs, enabling the organism to respond to its environment and maximize its survival chances [53]. Sleep affects cognitive restoration, emotional regulation, and overall health. Sleep architecture comprises rapid eye movement (REM) and non-REM sleep, forming 90 min sleep cycles that, in turn, contribute to overall sleep [54]. Factors affecting sleep quality include external factors, such as improper sleep hygiene, certain medications, or conscious disruption of the circadian rhythm in the form of jet lag and night-shift work, as well as internal factors, which comprise physical discomfort, stress, anxiety, and homeostatic imbalance duration [55]. Common sleep disorders include insomnia, obstructive sleep apnea, hypersomnia, or restless leg syndrome. According to DSM-5, insomnia is a sleep–wake disorder with a predominant complaint of significant insufficient sleep quality/quantity for at least 3 months, with at least one of the following symptoms: difficulty in falling asleep, difficulty in maintaining sleep, and early awakening with difficulties falling asleep again. Obstructive sleep apnea is a disorder characterized by repetitive upper-airway obstruction leading to hypopneas/apneas during sleep. The diagnosis of hypersomnia is based on chronic excessive daytime sleepiness for at least 3 months with concomitant adequate/prolonged nighttime sleep duration. Restless leg syndrome is a disorder with the presence of a severe urge to move legs accompanied by uncomfortable sensations with alleviations in leg movement. Maintaining an appropriate amount and quality of sleep can lower the risk of developing multiple disorders, as well as alleviate existing ones [56]. Common sleep disorders include insomnia, obstructive sleep apnea, hypersomnia, or restless leg syndrome.

People with ADHD are at a higher risk of developing sleep disturbances than the general population [57]; this may be because sleep disruption influences neurodevelopment in growing humans and also impairs neurocognitive functions, as mentioned before. Studies show that people with ADHD are approximately twice as likely to experience sleep problems as their healthy counterparts. According to research, children with ADHD experience more wake after sleep onset time and a longer latency of stage 3 of NREM sleep than their healthy counterparts [58]. One study showed that for subjects with ADHD, the REM phase was shorter, and a smaller percentage of time was spent in the REM phase in comparison to the control group [59]. The most prevalent problems among children include insomnia, night awakenings, and daytime sleepiness [60]. The most prevalent problems among adolescents are insomnia, restless leg syndrome, and frequent snoring [61]. The most prevalent problems in the adult population are night awakenings, daytime sleepiness, and psychosomatic symptoms during sleep onset [62]. Not only does the disorder itself affect sleep quality, the medications intended to treat ADHD, stimulants like methylphenidate (norepinephrine-dopamine reuptake inhibitor), and non-stimulants like atomoxetine (selective norepinephrine reuptake inhibitor) have been shown to influence various aspects of sleep. Stimulants, like methylphenidate (norepinephrine-dopamine reuptake inhibitor), and non-stimulants, like atomoxetine (selective norepinephrine reuptake inhibitor), have been shown to influence various aspects of sleep. However, some ADHD medications were reported to improve sleep quality and decrease nighttime awakenings, with greater effects for methylphenidate [63]. They tend to change sleep patterns, lowering the total REM and total NREM and increasing sleep-onset latency (these results refer to guanfacine (alpha-adrenergic agonist), methylphenidate, dasotraline (serotonin-norepinephrine-dopamine reuptake inhibitor), L-theanine (glutamate reuptake inhibitor and a competitive low-affinity glutamate receptor antagonist), and lisdexamfetamine (prodrug converting to dextroamphetamine—norepinephrine-dopamine reuptake inhibitor—and L-lysine). [64]. At the same time, atomoxetine intake did not cause any sleep pattern changes. Interestingly, atomoxetine has been successfully administered in combination with oxybutynin to patients with obstructive sleep apnea. A combination of noradrenergic and antimuscarinic signaling ameliorated the disorder’s severity by improving the responsiveness of the genioglossus muscle [65,66]. 

ADHD and its treatment directly affect sleep, and also significantly contribute to behaviors that are unfavorable to the quality of sleep. Patients with ADHD are more prone to become addicted, as the disorder drives risky behaviors. The results of a meta-analysis by Rohner et al. showed that more than twenty percent of substance use disorder patients manifest ADHD symptoms. It was reported that people with ADHD, after drinking alcohol, had poorer sleep than their peers without ADHD in the same setting. The same study examined the influence of cannabis, showing no significant change depending on ADHD presence [67]. Another addiction more prevalent among ADHD patients influencing sleep is smartphone and internet addiction, which increases the severity of insomnia, anxiety, and neuroticism in young adults [68,69]. For those with internet gaming disorders, ADHD seems to aggravate the ‘eveningness’ chronotype and insomnia [70], negatively affecting sleep quality. Food addiction is also common among ADHD individuals and is connected to the ‘eveningness’ chronotype. Additionally, ADHD is a risk factor for obesity [71]. A high-fat diet predisposing to obesity induces REM sleep fragmentation by dysregulating the dopaminergic signaling pathway [72,73].

Importantly, there is a bidirectional relationship between ADHD symptoms and sleep disturbances. Poor sleep quality, which can result from ADHD, in turn, worsens the ADHD symptoms, forcing behavioral consequences such as increased hyperactivity, impulsivity, inattention [74], cognitive effects, like impaired executive function, memory deficits, or reduced academic performance [75], and long-term health implications, including the increased risk of comorbid conditions such as depression or metabolic disorders [76,77]. Analogically, treating comorbid sleep problems can result in ameliorating ADHD symptoms; Fadeuilhe et al. showed that effects can be noticed after 6 months of treatment for comorbid insomnia disorder. Furthermore, the introduction of treatment for sleep disorder resulted in better ADHD outcomes, reducing its severity [78].

The intricate interplay between sleep, ADHD, and its bidirectional consequences appears to be inextricably linked and prone to change. The underlying mechanism involved in the connection has not been widely examined. However, immunological pathways may provide some insight into the workings of these interactions.

## 4. Mechanisms Linking Sleep and Immunology

### 4.1. Immunology of Healthy Sleep

The crosstalk between sleep and immunology occurs in health and disease. An adequate amount of sleep, along with its quality, provides support for both the non-specific immune response (cytokine production) and the specific immune response (T-cell function). Nighttime sleep, in comparison to wakefulness, reduces the number of monocytes, NK cells, and all lymphocyte subsets. At the same time, it increases the IL-2 production by T-cells but not IL-1beta, TNF-alpha, or IL-6 levels. These effects are seen to be independent of cortisol changes [79]. Sleep reduces T-cell amount but also affects the ratio—early nocturnal sleep imposes a change in T-cell helper (Th) types with enhanced Th1 activity, which, in later stages of sleep, is replaced by Th2 dominance [80]. Similarly, the complement system during sleep in healthy individuals is enhanced l [81]. These, and other changes that undergo profound alterations in response to the circadian cycle, help to regulate the immune system, preparing for a new day. After vaccination or during inflammation, sleep acts as a natural adjuvant protection and contributes to the strengthening of the immune response [82]. Sleep quality is associated with neurocognitive conditions via IL-18 and IL-12 [83]. In addition to regulating cytokines, proper sleep affects blood cell morphology. White blood cells and granulocytes were found to correlate negatively with sleep efficiency. In addition, the granulocyte level correlated negatively with REM latency [84].

### 4.2. Influence of Immune Impairment on Sleep

Excessive or prolonged immune activation due to impaired regulation or disease may deteriorate sleep structure and efficiency [85]. Even the most common general symptoms, including fatigue, fever, or pain, result in a decrease in short-term sleep quality and quantity. Prolonged symptoms induce more severe changes, such as chronic non-restorative sleep with altered sleep architecture and hypersomnia/insomnia, accompanied by excessive daytime sleepiness resulting from this [86,87,88]. In addition, many autoimmune diseases, such as rheumatoid arthritis [89], inflammatory bowel disease [90], multiple sclerosis [91], and psoriasis [92], seem to have a detrimental effect on sleep. One-third of patients with autoimmune diabetes were reported to have altered sleep quality, reflected as insufficient total sleep time [93].

### 4.3. Influence of Sleep Disturbances on the Immune System

Sleep deprivation induces unfavorable changes in the immune system, leading to increased susceptibility to infections and a chronic inflammatory state. Partial sleep deprivation results in temporarily reduced mitogen-induced cell proliferation, lower levels of HLA-DR expression, increased CD14 expression, and changes in CD4 and CD8 levels [94]. There is also a significant decrease in IL-6 levels [95]. Acute loss of sleep predisposes to cardiovascular diseases as it elevates the level of catecholamines [96]. Chronic sleep restriction, such as that caused by working night shifts, also exacerbates these changes [97]. It is notable that the changes seem to be reversible after recovery sleep or napping [98], but not entirely, as Th cells and IgA levels were reported not to be reestablished after sleep recovery [99]. Sleep deprivation decreases the melatonin level, which may contribute to further disruption of the circadian rhythm disruption [100]. Moreover, it participates in neurodegenerative disease development via oxidative stress exacerbation [101]. Sleep insufficiency is also a risk factor for gut microbiota dysbiosis [102]. Furthermore, sleep deprivation was reported to induce an elevated granulocyte count with no statistically significant changes in other parameters [84]. As presented, sleep generally acts as one of the primary regulators of the immune system; its quality influences the direction of immune development. Not only sleep but the entire circadian rhythm also affects human immunology, with interactions present in both directions. Thus, seeing that ADHD is connected with sleep disorders and immunology on multiple levels, and immunology is profoundly directed by sleep, one can look for some dependencies between ADHD and both sleep and immunology. The most important immune changes can be found in Table 2.

## 5. Relationship Between Sleep and Immunology in ADHD

Patients with ADHD are more likely to develop autoimmune diseases throughout life than patients without ADHD. The diseases include ankylosing spondylitis (OR = 2.78), ulcerative colitis (OR = 2.31), and autoimmune thyroid disease (OR = 2.53). There was also an association between ADHD and allergic diseases. Asthma (OR = 1.53), allergic rhinitis (OR = 1.59), atopic dermatitis (OR = 1.53), and urticaria (OR = 1.39) were significantly more prevalent among ADHD patients [103]. Patients with the dual diagnosis of ADHD and tic disorder were more likely to develop allergic diseases (OR = 1.73), including allergic rhinitis (co-occurrence 43% vs. 19.7%), asthma (27.5% vs. 11.9%), atopic dermatitis (10.6% vs. 5.9%), and allergic conjunctivitis (55.6% vs. 36.3%) [104]. In addition, a family history of several autoimmune diseases (thyrotoxicosis, type 1 diabetes, autoimmune hepatitis, psoriasis, ankylosing spondylitis) was connected to an increased risk of ADHD in offspring [105]. The maternal presence of autoimmune disease appears to be the most significant factor in the family history.

The exact connection between ADHD and immunological diseases has not yet been discovered; however, some connections suggest that sleep is associated with ADHD via immunology processes that act bidirectionally, depending on the stimuli. Understanding the role of the circadian rhythm may be crucial to comprehending these changes. The biological pathways can also include HPA axis dysregulation. Finally, consideration should be given to the role of neurotransmitters, such as dopamine and norepinephrine, in sleep and immune function among patients with ADHD. All these pathways exacerbate inflammation, potentially including neuroinflammation and oxidative stress.

The circadian rhythm is composed of four main genes and their proteins: circadian locomotor output cycles kaput (CLOCK); cryptochrome (CRY); basic helix–loop–helix ARNT-like protein (BMAL); and period circadian regulator (PER). These influence not only the sleep–wake cycle but 30% of all genes via binding to the E-box sequences present in gene promotors [106]. The circadian cycle was reported to be altered among ADHD subjects, with the most significant polymorphisms of gene CLOCK 3′UTR rs534654 [107] and rs1801260 [108]. Additionally, in ADHD patients, the rhythmicity of BMAL1 and PER2 was lost in comparison to healthy individuals [109]. ADHD is mainly a genetic-based disorder; thus, the genetic mutation of the circadian rhythm could predispose to a psychiatric phenotype, or vice versa. A regular level of BMAL1 is crucial for maintaining the second-level temporal intervals responsible for cognitive performance; thus, impaired circadian gene expression might predispose individuals to ADHD or worsen the clinical presentation of the disorder. Animal models showed that a proper CRY protein inhibits proinflammatory cytokine expression, including TNF and IL-16, which may have a protective effect on neurodevelopment.

ADHD patients are reported to have significantly higher TNF-alpha levels in comparison to their healthy counterparts [110]. Differences in CRY1 expression are seen in untreated ADHD patients compared to controls and ADHD patients with medication [111]. Children with ADHD present with increased IL-16, which is associated with hyperactivity [112]. Another study connected IL-16 in ADHD with poorer infant health [113]. An altered variant of the CRY gene present among some ADHD patients may influence IL-16 and TNF-alpha levels, causing more severe hyperactive–impulsive symptoms. The deficiency of CRY protein may relieve its inhibitory effect on cyclic adenosine monophosphate (cAMP) production, leading to elevated cAMP levels and enhanced protein kinase A activation, which further promotes the nuclear factor kappa-light-chain-enhancer of activated B cells activation via phosphorylation of transcription factor p65. Therefore, a disruption in circadian rhythms contributes to greater vulnerability to chronic inflammatory diseases [114]. In addition, CRY proteins were shown to regulate autoimmunity, as its deficiency affects B-cell development, the B-cell receptor signaling pathway, and component C1q expression [115]. It may have an application, as defective CRY places an individual at a greater risk of developing autoimmune disease, which is associated with an increased risk of ADHD [105].

HPA axis dysregulation, which manifests in abnormal cortisol levels, might be another component in the association between sleep, ADHD, and the immune system. Patients were reported to have delayed phases of cortisol rhythms [109]. The most prominent changes may occur in the predominantly hyperactive–impulsive ADHD type, where the levels are the lowest [116]. An effective treatment after 6 months was shown to improve cortisol level disruption [117]. As cortisol secretion follows a circadian rhythm, its dysfunction can be connected to sleep–wake dysregulation. Interestingly, such a delayed circadian phase can be treated with bright light therapy to normalize melatonin’s nocturnal rise and cortisol’s morning rise in people with ADHD symptoms [118]. Previously, bright light therapy was instituted for the prevention of comorbid depression development among ADHD patients, with good results [119]. The HPA axis was found to be a modulator between qualitative and quantitative sleep and immune changes, with these exerting a bidirectional influence [120]. The majority of sleep problems that affect ADHD patients have been connected to HPA axis alternations [120,121,122]. Sleep duration influences proinflammatory cytokines, especially IL-6 and TNF-alpha [123]. The association between immune markers (IL-6 and TNF-alpha) and the cortisol awakening response in ADHD may vary by subtype—specifically, the inattentive subtype showed a negative correlation.

In contrast, the combined type was not statistically significant [124]. Some genetic predispositions should be taken into consideration, as glucocorticoid receptor NR3C1 variants among ADHD children were reported to have a moderate influence on ADHD severity and the occurrence of concurrent conduct disorder [23]. One of these variants—rs6189—has been associated with circulating IL-6 levels and is linked to sleep duration among healthy individuals, which may increase their inflammatory state [125]. Additionally, in one animal study, receptor dysregulation was found to be involved in ADHD development [126]. As both IL-6 and TNF-alpha can regulate the HPA axis at hypothalamic, pituitary, and adrenal levels [122,126,127], its sleep-origin stimulation may result in worsening the clinical picture of ADHD patients. Additionally, HPA dysregulation primarily due to ADHD may worsen the circadian rhythm and sleep quality. Some other immune markers have been examined, such as salivary alpha-amylase and secreted immunoglobulins (IgA, IgG, and IgM); however, these did not show any association with cortisol.

Catecholamines, such as adrenaline, noradrenaline, and dopamine, through their modulatory effect on fronto-striato-cerebellar circuits, typically affected in ADHD, have been the target of the majority of medications used to alleviate ADHD symptoms [128]. The pharmacologic aim of both psychostimulants and non-psychostimulants is to increase central dopamine and noradrenaline activity in brain circuits [129]. As a disturbed monoaminergic system is associated with ADHD, it may influence sleep as well. Catecholamines are mainly involved in the wake-promoting state throughout the day [130]. The aforementioned circadian clock genes (especially BMAL1, NPAS2, and PER2) can affect the transcription of monoamine oxidase A, which is responsible for monoamine degradation [131]. Thus, any of the disturbances in circadian clock components may aggravate ADHD symptoms, comorbid aggression, and sleep disturbances [132]. On the other hand, particular gene polymorphisms, such as catechol-O-methyltransferase, which increase the risk of ADHD, are also involved in modulating sleep quality, which might make this group of patients susceptible to developing sleep disturbances [133]. Untreated, sleep disorders can jointly affect the immune system, which is dependent on the catecholamines in responding to ongoing infections; both sleep insufficiency [134] and ADHD [135] have been recognized as risk factors for infection. However, there are no studies describing the potential influence of an impaired catecholamine system on the immune system concerning the shown sleep parameters.

In addition to catecholamine dysregulation, monoamine neurotransmitter disorders should be mentioned as a possible bridge between ADHD, sleep disorders, and the immune system. Monoamine neurotransmitter disorders, although a rare group of inherited disorders, involve imbalances in dopamine, noradrenaline, and serotonin, which can directly and separately affect ADHD symptoms [136], sleep quality [137], and the immune system [138] via changed transmitter levels. However, to date, no studies have researched this topic as a whole; thus, further research is needed to assess predictable connections and the effect of monoamine oxidase inhibitors on sleep and immunity among ADHD patients.

As ADHD patients constitute a heterogeneous group and the mechanisms responsible for the development of the disorder are not strictly defined, it seems unlikely that a universal connection between ADHD, sleep, and the immune system will be found. However, based on the most prevalent aspects described in the literature, some significant relationships emerge. A summary of exact associations is presented in Figure 1.

## 6. Therapeutic Interventions

Proper treatment alleviates ADHD symptoms and most sleep disturbances, which, in effect, can improve immune health. There is a wide choice of suggested recommendations, depending on the patient’s characteristics. The most prevalent ADHD treatment includes behavioral interventions (cognitive–behavioral therapy (CBT), pharmacological treatments (sleep aids, melatonin, modifications of ADHD medication regimens), sleep hygiene practices, and lifestyle modifications (diet, exercise, stress management).

A study by Efron et al. reports that the most popular sleep medication among children with ADHD seems to be clonidine—implemented in 14% of young ADHD patients under examination, while melatonin was used in 9% of the study population. In the same group of patients, the intake of ADHD medication was 81% among children [139]. The same research shows that among adult ADHD individuals, approximately 47.5% reported sleep medication use in comparison to 12% from the control group. The most frequent were non-benzodiazepine drugs (zopiclone, zolpidem), followed by melatonin and antihistamines (propiomazine) [57]. In addition to typical sleep medications, modifying ADHD medication regimens, depending on the case, seemed to be efficient in improving sleep quality. As stimulants were shown to disturb sleep architecture, it is reasonable to consider replacing medication with non-stimulants if sleep complaints occur [64].

The incorporation of behavioral interventions in ADHD is crucial for favorable outcomes, with psychotherapy also being a primary treatment of insomnia. CBT is especially recommended for both disorders, combined, when necessary, with medical treatment [140]. Even a short-term behavioral sleep intervention was reported to moderately improve the severity of ADHD symptoms [141]. CBT therapy can be efficacious, particularly in insomnia treatment; a 3-month treatment duration was shown to improve the Insomnia Severity Index by 6.8 points compared to a pre-treatment evaluation [142]. Not only does undergoing therapy have a significant effect, similar results can also be achieved by adhering to proper sleep hygiene. Poorer sleep hygiene has been associated with subjective sleep problems, including increased sleep latency, trouble resuming sleep, and difficulty waking up [143]. Psychoeducation on sleep hygiene is a low-cost strategy that can reduce complaints and support families with an ADHD member [144].

Finally, long-term lifestyle modifications, such as diet, exercise, or stress management, were reported to make a significant change in ADHD sleep improvement [145]. Patients with poorer dietary habits—higher intake of carbohydrates, fats, and, most particularly, sugar—were more prone to experience sleep disturbances and nighttime sweating [146]. After proper diet application, complaints of headaches or stomachache, increased thirst or perspiration, and poor sleep quality were reduced [147]. Furthermore, among children with ADHD, physical activity has been reported to improve sleep efficiency, sleep onset latency, and wake after sleep onset [148]. Among children, moderate-to-vigorous physical activity was negatively associated with sleep latency and positively related to working memory, a component of executive function outcomes [149]. Among adults on stimulant medication, moderate-to-vigorous physical activity improved sleep quality; however, this was observed only in men. With at least 105 min of moderate-to-vigorous physical activity, the incidence of sleep troubles gradually decreased; 341 min was shown to make a significant difference [150].

However, the mentioned therapeutic interventions have their cons, including delayed response to treatment, especially in non-pharmacological treatments, such as CBT and sleep hygiene/lifestyle changes, as a result of the reduced general attention span, inconsistent routines, and executive dysfunction typical in ADHD subjects. Side effects, such as mood changes and a risk of dependence, are common in pharmacological treatments. In addition, these may interact with other medications used. Finally, while using any option—pharmacological and non-pharmacological—one should not overlook other comorbidities, as there can be a variability in effectiveness. Keeping these limitations in mind may contribute to a more personalized approach and a better response to the implemented therapy. All treatment options comprise more sleep-focused interventions than immune interventions, as there is a gap in evidence directly linked to immune-modulatory therapies and their effect on sleep and ADHD symptoms. Although currently inconclusive and in need of more research, hypotheses include the influence of classic anti-inflammatory agents on sleep and ADHD, and the influence of probiotic supplementation on both sleep and ADHD.

The treatment plan for each ADHD case should be considered individually, tailored to the needs of each individual, in accordance with the latest recommendations. The best results are mainly achieved by combining properly chosen ADHD medications and non-pharmacologic interventions. Every change in the treatment plan should be clearly explained and discussed with the patient, as it may be a challenge to fulfill the recommendations. A summary of this information is depicted in Figure 2.

## 7. Conclusions

This review describes the intricate and bidirectional relationships between sleep disturbances and immunological dysregulation in patients with ADHD. The disorder is increasingly thought to have systemic physiological implications, most notably involving alterations in sleep architecture and immune function. Sleep disturbances are highly prevalent among individuals with ADHD, affecting up to 50% of patients, and are characterized by increased sleep-onset latency, reduced REM sleep, and a greater number of nocturnal awakenings. Notably, the relationship appears bidirectional, with ADHD symptoms themselves further disrupting sleep patterns, thus making it a self-perpetuating cycle. The immunological findings presented in this article reveal a consistent pattern of low-grade systemic inflammation in individuals with ADHD, characterized by elevated concentrations of cytokines, such as IL-6 and TNF-α, dysregulation of the hypothalamic–pituitary–adrenal (HPA) axis, and genetic variations that affect immune response pathways. Disruptions in circadian rhythm genes—CLOCK, BMAL, PER, CRY—implicate the central role of clock gene regulation in linking sleep, immune function, and ADHD pathophysiology. Additionally, catecholaminergic dysfunction, central to ADHD neurobiology, influences both arousal states and immune modulation. The cumulative effects of genetic predisposition, prenatal and early-life environmental exposures (e.g., maternal inflammation, infection, air pollution), and lifestyle factors (e.g., diet, substance use, circadian misalignment) significantly shape the clinical phenotype and progression of ADHD. Emerging evidence implicates immune-mediated neuroinflammation and oxidative stress as key processes that may bridge the gap between disturbed sleep and ADHD symptomatology. Therapeutic strategies targeting both sleep and immune regulation offer promising avenues for improving ADHD management. The pharmacological modulation of sleep, careful titration and selection of ADHD medications to minimize sleep disruption, cognitive–behavioral interventions, and lifestyle modifications centered around sleep hygiene, dietary optimization, and physical activity represent critical components of a comprehensive treatment framework. Moreover, early identification and intervention in sleep and immune dysfunctions may serve as modifiable risk factors to improve long-term outcomes.

In summary, ADHD should be conceptualized not just as a neurodevelopmental disorder but rather as a systemic condition involving profound relationships between neurocognitive, immunological, and sleep regulatory systems. Future research should prioritize longitudinal, mechanistic studies to elucidate these connections further and inform the development of individualized, multimodal therapeutic approaches aimed at improving clinical trajectories and overall quality of life in patients with ADHD.

## Figures and Tables

**Figure 1 ijms-26-07967-f001:**
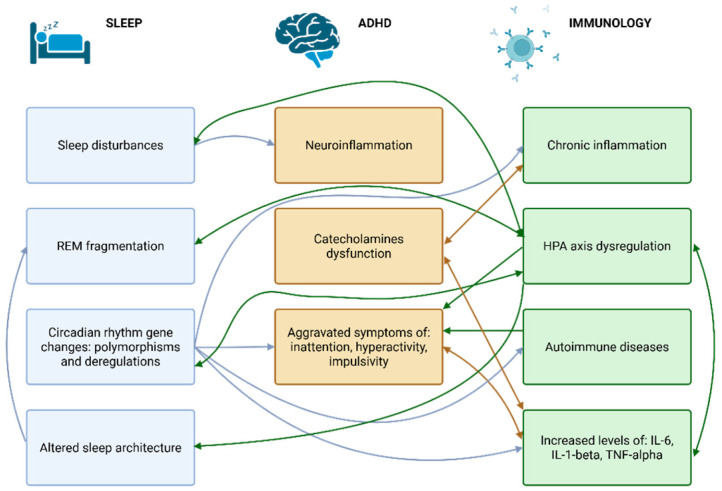
The associations between sleep, ADHD, and immunology. Abbreviations: ADHD—attention deficit hyperactivity disorder; HPA—hypothalamic–pituitary axis; IL—interleukin; REM—rapid eye movement; TNF—tumor necrosis factor.

**Figure 2 ijms-26-07967-f002:**
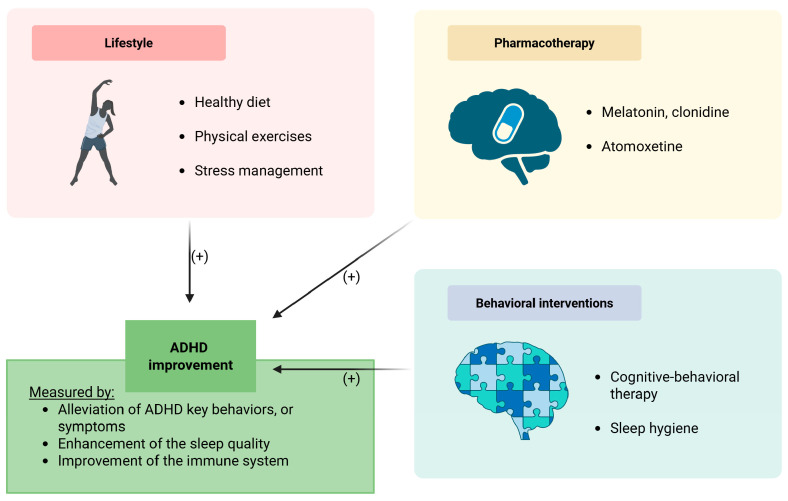
Therapeutic interventions among patients with ADHD. Abbreviations: ADHD—attention deficit hyperactivity disorder. Symbol (+) means a positive contribution.

**Table 1 ijms-26-07967-t001:** Immunological aspects of ADHD.

Immunological Aspects of ADHD		
Category	Description	Citation
Increased proinflammatory cytokines	IL-6, IL-1-beta, and TNF-alpha increased in blood; markers correlate with the severity of the disease	[17]
HPA axis dysregulation	Decreased morning cortisol level and decreased daily cortisol level in blood	[20]
Increased proinflammatory salivary markers	Alpha-amylase, IgA, and IgM increased in saliva; no differences in morning cortisol in saliva	[21]
Blood morphology changes	Increased dendritic cells, monocytes, and granulocytes	[19]
Genetics polymorphisms	Polymorphisms of FKBP5, NR3C1, DAT1, DRD4, and 5-HTTLPR	[22,23,28,29,30]
Changed microbiota	Gut microbiota imbalance changed potentially via stimulant usage and a higher need for antibiotic usage	[40,42]
Increased prevalence of autoimmune diseases	Asthma, diabetes mellitus type 1, celiac disease, allergies, and thyroid abnormalities are more prevalent among ADHD subjects in comparison to non-ADHD subjects	[43,45,47,49,52]

Abbreviations: 5-HTTLPR—serotonin-transporter-linked promoter region, ADHD—attention deficit hyperactivity disorder, BMAL1—basic helix-loop-helix ARNT-like protein 1, CLOCK—circadian locomotor output cycles kaput, CRY1—cryptochrome circadian regulator 1, DAT1—dopamine transporter 1, DRD4—dopamine receptor D4, FKBP5—FK506 binding protein 5, HPA—hypothalamic–pituitary axis, Ig—immunoglobulin, IL—interleukin, NR3C1—glucocorticoid receptor, PER2—period circadian regulator 2, TNF—tumor necrosis factor.

**Table 2 ijms-26-07967-t002:** Immune change in healthy sleep vs. sleep insufficiency.

Immune Change in Healthy Sleep vs. Sleep Insufficiency		
Category	Healthy Sleep	Sleep Insufficiency
Cytokines	Affect within physiological limits with a marked increase in the IL-2, IL-12, and IL-18	A decrease in the IL-6
Cells	Affect within physiological limits with a marked decrease in the granulocytes, NK cells, and all lymphocyte subsets, with first Th1 and then Th2 dominance	An increase in granulocytes
MHC	Affect within physiological limits	Lower HLA-DR expression
Gut microbiota	Affect within physiological limits	Dysbiosis in long-term
Melatonin	Affect within physiological limits	A decrease in the melatonin levels
Stress hormones	Affect within physiological limits	An increase in the level of catecholamines

Abbreviations: IL—interleukin, MHC—major histocompatibility complex, NK—natural killer, Th—T-helper.

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
