# Peer review of "Relationship Between Sleep and Immunology in Attention Deficit Hyperactivity Disorder"

_ijms, 2025, doi:10.3390/ijms26167967_

Round 1
Reviewer 1 Report
Comments and Suggestions for Authors
The authors conducted a comprehensive review of the literature on the relationship between sleep disorders and changes in the immune system during the prenatal and postnatal periods of development, as well as the role of these changes in the pathogenesis of ADHD. The authors rightly point out that ADHD ought to be understood not only as a neurodevelopmental disorder but also as a systemic condition. This is important given the increasing number of adults being diagnosed with ADHD worldwide. Sleep disorders are prevalent among patients with ADHD, accounting for approximately half of diagnosed cases. Not only are sleep disorders caused by ADHD, they also exacerbate its symptoms. The most interesting section of the review concerns the relationship between ADHD and the immune system, since immune factors are active before and after birth.
The authors also mention the role of monoaminergic systems in the brain in the pathogenesis of ADHD. They cite studies in which polymorphisms in the dopamine transporter (DAT1) and dopamine receptor (D4) genes correlate with ADHD symptoms. It would be interesting to hear the authors' opinion on whether common mechanisms underlie sleep disorders, the immune system and the pathogenesis of ADHD in case of monoamine metabolism disorders. This aspect should be reflected in the relevant section of the review.
Author Response
Author's Reply to the Review Report (Reviewer 1)
Thank You very much for taking the time to review this manuscript. Please find the detailed responses
below and the corrections highlighted in the re-submitted files.
Comment:
“The authors conducted a comprehensive review of the literature on the relationship between sleep
disorders and changes in the immune system during the prenatal and postnatal periods of development,
as well as the role of these changes in the pathogenesis of ADHD. The authors rightly point out that
ADHD ought to be understood not only as a neurodevelopmental disorder but also as a systemic
condition. This is important given the increasing number of adults being diagnosed with ADHD
worldwide. Sleep disorders are prevalent among patients with ADHD, accounting for approximately half
of diagnosed cases. Not only are sleep disorders caused by ADHD, they also exacerbate its symptoms.
The most interesting section of the review concerns the relationship between ADHD and the immune
system, since immune factors are active before and after birth. The authors also mention the role of
monoaminergic systems in the brain in the pathogenesis of ADHD. They cite studies in which
polymorphisms in the dopamine transporter (DAT1) and dopamine receptor (D4) genes correlate with
ADHD symptoms. It would be interesting to hear the authors' opinion on whether common
mechanisms underlie sleep disorders, the immune system and the pathogenesis of ADHD in case of
monoamine metabolism disorders. This aspect should be reflected in the relevant section of the
review.”
Response:
Thank You so much for the suggestion concerning adding the monoamine metabolism disorders topic
as a bridge between ADHD, the immune system, and sleep disorders. Although not yet discovered, it
seems a promising and essential connection to investigate. Please find the corrected version of the
fragment below.
Catecholamines such as adrenaline, noradrenaline, and dopamine, through their modulatory
effect on fronto-striato-cerebellar circuits, typically affected in ADHD, have been the target of the
majority of medications used to alleviate ADHD symptoms [131]. The pharmacologic aim of both
psychostimulants and non-psychostimulants is to increase central dopamine and noradrenaline activity
in brain circuits [132]. As a disturbed monoaminergic system is associated with ADHD, it may influence
sleep as well. Catecholamines are mainly involved in the wake-promoting state throughout the day
[133]. The aforementioned circadian clock genes (especially BMAL1, NPAS2, and PER2) can affect the
transcription of monoamine oxidase A, which is responsible for monoamine degradation [134]. Thus,
any of the disturbances in circadian clock components may aggravate ADHD symptoms, comorbid
aggression, and sleep disturbances [135]. On the other hand, particular gene polymorphisms, such as
catechol-O-methyltransferase, which increase the risk of ADHD, are also involved in modulating sleep
quality, which might make this group of patients susceptible to developing sleep disturbances [136].
Untreated, sleep disorders can jointly affect the immune system, which is dependent on the
catecholamines in responding to ongoing infections; both sleep insufficiency itself [137] and ADHD
itself [138] have been recognized as risk factors for infection. However, there are no studies describing
the potential influence of an impaired catecholamine system on the immune system concerning the
shown sleep parameters.
In addition to catecholamine dysregulation, monoamine neurotransmitter disorders should
be mentioned as a possible bridge between ADHD, sleep disorders, and the immune system.
Monoamine neurotransmitter disorders, although a rare group of inherited disorders, involve
imbalances in dopamine, noradrenaline, and serotonin, which can directly affect separately ADHD
symptoms [139], sleep quality [140], and the immune system [141] via changed transmitter levels.
However, to date, no studies have researched this topic as a whole; thus, it needs exploration to assess
predictable connections and the effect of monoamine oxidase inhibitors on sleep and immunity
among ADHD patients.
As ADHD patients constitute a heterogeneous group and the mechanisms responsible for the
development of the disorder are not strictly defined, it seems unlikely to find a universal connection
between ADHD, sleep, and the immune system. However, based on the most prevalent aspects
described in the research, some main relationships emerge. The summary of exact associations from this
paragraph is presented in Figure 1.

Reviewer 2 Report
Comments and Suggestions for Authors
The manuscript provides a comprehensive and timely review of the emerging interconnection between sleep disturbances, immune dysfunction, and ADHD. It covers a wide breadth of literature spanning neurodevelopmental, immunological, circadian, and behavioral perspectives. The central thesis, that immunological dysregulation and sleep disturbances are both implicated in ADHD pathophysiology and may share overlapping mechanisms, is clearly articulated. However, the manuscript suffers from several critical issues related to scientific rigor, writing clarity, redundancy, and lack of synthesis. Below are section-by-section comments:
- The abstract is somewhat verbose and lacks structural clarity. Improve conciseness and structure by clearly segmenting background, rationale, findings, and conclusion in the abstract. Specify key mechanistic insights or unique contributions of this review.
- Provide a succinct rationale for integrating immunology and sleep studies in the context of ADHD.
- In section 2, Immunological Aspects of ADHD: Streamline overlapping sections and integrate mechanistic interpretation. Focus more on how immune dysfunction might causally contribute to ADHD phenotypes rather than listing associations. Avoid citation stacking without clear analytical commentary.
- In section 3, Importance of sleep and sleep disruption in ADHD: Highly repetitive; underdeveloped mechanistic discussion, how exactly do sleep disruptions interface with neurodevelopment?; More rigorously define types of sleep disorders and their diagnostic criteria.
- In section 4, Mechanisms linking sleep and immunology: Lacks cohesion; paragraph flow is disjointed with abrupt topic switches; Overlaps with previous section without clear delineation; Include a concise summary table of immune changes under sleep sufficiency vs. deprivation.
- In section 5, Relationship between sleep and immunology in ADHD: Provide a clear conceptual model illustrating how circadian gene dysfunction, immune activation, and ADHD symptoms intersect (beyond Figure 1, which is too general); Reduce the exhaustive listing of genes and SNPs unless mechanistically relevant to the central argument.
- In section 6, Therapeutic Interventions: Discuss limitations of current treatments and highlight gaps in evidence linking immune-modulatory therapies to ADHD; Suggest testable hypotheses or future clinical trial designs targeting sleep-immune-ADHD nexus.
- While generally informative, the manuscript contains multiple grammatical inconsistencies, awkward phrasing, and typographical errors. A thorough professional language edit is strongly recommended.
Need improvement
Author Response
Author's Reply to the Review Report (Reviewer 2)
Thank you sincerely for dedicating Your time to reviewing this manuscript. Below, you will find our
detailed responses, along with the corresponding revisions and corrections marked in the re-submitted
files.
Comment 1:
“The manuscript provides a comprehensive and timely review of the emerging interconnection between
sleep disturbances, immune dysfunction, and ADHD. It covers a wide breadth of literature spanning
neurodevelopmental, immunological, circadian, and behavioral perspectives. The central thesis, that
immunological dysregulation and sleep disturbances are both implicated in ADHD pathophysiology and
may share overlapping mechanisms, is clearly articulated. However, the manuscript suffers from several
critical issues related to scientific rigor, writing clarity, redundancy, and lack of synthesis. Below are
section-by-section comments:
The abstract is somewhat verbose and lacks structural clarity. Improve conciseness and structure by
clearly segmenting background, rationale, findings, and conclusion in the abstract. Specify key
mechanistic insights or unique contributions of this review.
Provide a succinct rationale for integrating immunology and sleep studies in the context of ADHD.”
Response 1:
Thank You for the notice. While writing the aforementioned abstract, we tried to provide a complex
but short abstract with the most important information included inside in the order in the text.
However, for clearer perception, we added a segmented structure as You recommended.
Background: Attention-deficit/hyperactivity disorder (ADHD) is a complex neurodevelopmental
disorder that not only affects attention and behavior but is also intricately linked with sleep
disturbances and immune system dysregulation.
Rationale: Recent research highlights that individuals with ADHD frequently experience sleep
problems, which in turn exacerbate ADHD symptoms and contribute to cognitive and emotional
difficulties. Immunological alterations, including elevated proinflammatory cytokines and
hypothalamic-pituitary-adrenal axis dysfunction, have been observed among ADHD patients,
suggesting a biological interplay between inflammation, sleep, and neurodevelopment. Genetic and
environmental factors further modulate these relationships, influencing the onset and progression
of the disorder. Thus, there is a need to find a key connecting such topics and most vulnerable
subjects to contribute to more personalized approach.
Findings: This review examines the complex relationships between sleep, immunology, and ADHD,
exploring the underlying mechanisms that involve circadian rhythm genes, neuroinflammation, and
neurotransmitter imbalances.
Conclusion: this review outlines therapeutic strategies, emphasizing the importance of integrated
pharmacological, behavioral, and lifestyle interventions to improve sleep quality, regulate immune
responses, and ultimately enhance the overall management of ADHD.
Comment 2:
“In section 2, Immunological Aspects of ADHD: Streamline overlapping sections and integrate
mechanistic interpretation. Focus more on how immune dysfunction might causally contribute to
ADHD phenotypes rather than listing associations. Avoid citation stacking without clear analytical
commentary.”
Response 2:
Thank You for the comment. We didn’t divide the paragraph into sections, as for us it made more sense
as a whole to show how interdependent and intricate the connection is. Additionally, we created a
table with the most important information. However, thanks to the advice, we put the paragraph in
order as recommended. Also, we added commentaries in some citations and provided more text about
ADHD phenotypes – here the information is not well established in my opinion, that is why I didn’t
mention such a connection – and I hope it is clearer now.
2.1 General overview of immune changes among ADHD subjects
In recent years, more research has explored the association between ADHD and the
immunological aspects. In general, people with ADHD have altered immunological profiles, including
typically higher levels of interleukin 6 (IL-6; known to influence microglia), IL-1-beta (a key
proinflammatory cytokine); known to influence microglia) [14], IL-1-beta (a key proinflammatory
cytokine), and tumor necrosis factor (TNF) alpha (related to a weak immune response), which
negatively correlate with the symptom severity [17]. It means ADHD subjects are more prone to
developing immune disorders due to cytokine disruption. MicroRNA-200b-3p or taurine intake seems
to lower those cytokines [18], which may be a promising pathway to search for a treatment target. As
ADHD is perceived as a neurodevelopmental disorder without a strictly defined genesis, the evaluation
of particular microRNAs might be helpful in unraveling those biochemical relations. At a molecular
level, ADHD contributes to the increase of myeloid dendritic cells, monocytes, and granulocytes. In the
saliva of ADHD subjects, there were significantly elevated levels of alpha-amylase, immunoglobulin A,
and immunoglobulin M, but no differences in morning cortisol were detected [19]. In the saliva of
ADHD subjects, there were significantly elevated levels of alpha-amylase, immunoglobulin A, and
immunoglobulin M, but no differences in morning cortisol were detected. However, according to a
systematic review by Chang et al., ADHD individuals have lower cumulative cortisol with lower
morning levels [20]. It means stress markers seem to be increased among ADHD subjects, pointing
to a plausible role of such enhanced immune response. Besides, children with ADHD are found to
have down-regulated hypothalamic-pituitary-adrenal (HPA) axis as showed in the presence of single
nucleotide polymorphisms (especially rs9470080) in FK506 binding protein 5 – involved in
immunoregulation via, e.g., interacting with corticoid receptor complexes – and lower diurnal cortisol
levels in comparison to healthy children [21]. In addition, glucocorticoid receptors TthIIII_rs10052957
[22]. These signs of subclinical inflammation in morphology, cytokine levels, or hormone disbalances
might be connected with the etiology of ADHD, which so far has not been explained. [23].
Both genetic and environmental factors contribute to shaping this profile. Genetic, as offspring
of ADHD parents are more prone to develop psychiatric disorders, including ADHD – up to 88% for
the formal heritability [24] – up to 74% for the formal heritability. Environmental, as many risk factors
have been suggested and confirmed to be associated with ADHD onset, e.g. maternal pre-pregnancy
obesity, childhood eczema, hypertensive disorders during pregnancy and pre-eclampsia, maternal
acetaminophen exposure during pregnancy, maternal smoking during pregnancy, childhood asthma,
maternal pre-pregnancy overweight, and serum vitamin D [25]. The influence of the mother's
immunology was connected especially in the context of the disorder's predisposition as well as the
offspring's immunology as a triggering factor of ADHD occurrence. Environmental factors, such as
increased air pollution, suggest that economically disadvantaged populations might be predisposed to
the syndrome [26]. Although many potential connections are not well-established, and the human
immune system remains incompletely understood, some preliminary conclusions can be drawn [27].
2.2 Impaired immunity before birth and ADHD symptoms
To begin with, differences can occur in the prenatal development of the immune system
between subjects who will later develop neurodevelopmental disorders and those who will not.
Although the heritability of ADHD is high, there are no confirmed genes, including proinflammatory,
responsible for this. The most promising by far are those connected with dopamine and serotonin
circuits. One polymorphism of dopamine transporter was proved to be associated with almost 3 times
greater prevalence of ADHD compared to children without this allele. Dopamine receptor D4
polymorphism presence contributed to increased risk of ADHD development as well. Both of the
polymorphisms can be the effect of inheritance but also prenatal smoking exposure. Serotonintransporter-linked polymorphic region (5-HTTLPR) [28]. Serotonin-transporter-linked polymorphic
region (5-HTTLPR) polymorphism was found to be the bridge between maternal anxiety and negative
emotionality, enhancing the vulnerability of the fetus to environmental risk factors for ADHD. Through
the same polymorphism, maternal smoking might affect the fetal serotonin system, causing the fetus'
vulnerability to emotional problems [29], [30], [31]. Insights into the genetic risk of developing ADHD
can be enhanced by the evaluation of pregnancy history. It means both groups of reasons cause innate
disruption. Mothers with diverse inflammatory factors (such as obesity, autoimmune disease, infection
during pregnancy, prolonged stress [32], smoking, and sleep disorders [33], and sleep disorders) are
more likely to give birth to a child with subsequent ADHD. This process probably occurs among others
through the activation of toll-like receptors [34], similar to other autoimmune disorders. In prenatal
offspring, it induces neuroinflammation affecting brain development and synaptic transmission, which
is consistent with the neuroinflammation hypothesis in ADHD pathophysiology, suggesting that
inflammation of nervous tissue can have a detrimental effect on the development of psychiatric
disorders [35]. As the nervous system starts its growth in the 3-4th week after conception, any
unfavorable factor affecting neurogenesis and subsequent development from this time has its
consequences in post-natal life. What is interesting is that similar changes occur in children and adults
with ADHD, and the extent of the morphologic changes correlates with specific subtypes of the disorder
in ADHD children, with more detrimental changes among unmedicated ones (cerebral volume lower
by 5.8%, cerebellar volume lower by 6.2%) based on magnetic resonance images [37]. What was
observed in the animal model, the application of anti-inflammatory treatment to the mother ameliorated
abnormal locomotor activity, indicating protection of neurodevelopment and creating an opportunity
to lessen the risk of ADHD occurrence in situations of excess maternal stress [38], [39].
2.3 Impaired immunity after birth and ADHD symptoms
After labor, detrimental immunology-affecting factors can also predispose one to develop
ADHD, with the best-described gut microbiome influence. A longitudinal study of over 16,000
Swedish children revealed that infant exposure to infectious agents and antibiotic usage, which affect
the human microbiome, is related to a higher risk of ADHD. Gut microbiome imbalance in later life
among people with ADHD was reported to be amended. However, in older patients, the preference for
a non-healthy diet can be the reason for such a connection. What is interesting is that a dietary
intervention in the form of vitamin D and vitamin D with magnesium as co-adjuvant treatment
improved ADHD pharmacological treatment [40]. Gut microbiome imbalance in later life among people
with ADHD was reported to be amended. However, in older patients, the preference for a non-healthy
diet can be the reason for such a connection. What is interesting is that a dietary intervention in the form
of vitamin D and vitamin D with magnesium as co-adjuvant treatment improved ADHD
pharmacological treatment. Influence on the gut microbiome is also exerted by psychostimulant
medications, pointing to differences in taxonomic beta diversity and lower evenness [41]. Influence on
the gut microbiome is also exerted by psychostimulant medications, pointing to differences in
taxonomic beta diversity and lower evenness. In addition, coexisting diseases derived from the
dysregulated immune system are more commonly present among ADHD patients than the rest of the
population, with the most popular including asthma [42]. In addition, coexisting diseases derived from
the dysregulated immune system are more commonly present among ADHD patients than the rest of
the population, with the most popular including asthma and autoimmunological diseases: allergy [43],
[44] and autoimmunological diseases: allergy, coeliac disease [45], [46], coeliac disease [47], [48],
diabetes mellitus type 1, diabetes mellitus type 1 [49], [50], or thyroid abnormalities. Although many of
the precise connections are still unknown, it can be postulated that these disorders work together to
create a self-perpetuating cycle that leads to a dysregulated immune system. [51], [52]. Although many
of the precise connections are still unknown, it can be postulated that these disorders work together to
create a self-perpetuating cycle that leads to a dysregulated immune system.
2.4 Conclusions of the paragraph
As described above, subjects with ADHD have an altered immunological system because of
both genetic and environmental factors interpenetrating each other. Therefore, the strongest scientific
risk factors should be taken into consideration while assessing the patient. The important thing to
mention is that the majority of studies did not stipulate reliable information on which ADHD
phenotypes the factor affects the most. As the awareness of risk factors is still not widespread,
preventive knowledge and actions should be expanded. The summary of this paragraph is shown in
Table 1.
Comment 3:
“In section 3, Importance of sleep and sleep disruption in ADHD: Highly repetitive; underdeveloped
mechanistic discussion, how exactly do sleep disruptions interface with neurodevelopment?; More
rigorously define types of sleep disorders and their diagnostic criteria.”
Response 3:
Thank You for the comment. The aim of section 3 was to provide reliable and condensed information
for the context in the following section. Based on the information available to date in research bases,
we tried to focus on the most common troubles without individual cases, which often do not repeat in
other studies. We improved our text with a description of sleep disorders, their diagnostic criteria, and
the neurodevelopment topic. The discussion is based on well-organized and easy to follow structure
making it more accessible for the reader.
Sleep is one of the most critical factors determining the quality of life, as its normal duration
takes around 1/3 of the day. Sleep is a recovery state of the sleep-wake cycle; thus, the nervous
system's development largely relies on it. Thanks to the circadian rhythm, a 24-hour oscillation occurs,
enabling the organism to respond to its environment and maximize its survival chances. Sleep is one of
the most critical factors determining the quality of life, as its normal duration takes around 1/3 of the
day. Thanks to the circadian rhythm, a 24-hour oscillation occurs, enabling the organism to respond to
its environment and maximize its survival chances [53]. What is important, sleep affects cognitive
restoration, emotional regulation, and overall health. Sleep architecture comprises rapid eye movement
(REM) and non-REM sleep, forming 90-minute sleep cycles that, in turn, contribute to overall sleep [54].
Factors affecting sleep quality include external factors such as improper sleep hygiene, certain
medications, or conscious disruption of the circadian rhythm in the form of jet lag and night-shift work,
as well as internal factors, which comprise physical discomfort, stress, anxiety, and homeostatic
imbalance duration [55]. Common sleep disorders include insomnia, obstructive sleep apnea,
hypersomnia, and restless leg syndrome. According to DSM-5, insomnia is a sleep-wake disorder with
a predominant complaint of significant insufficient sleep quality/quantity for at least 3 months with
at least one symptom from: difficulty in falling asleep, maintaining sleep, early awakening with
difficulties falling asleep again. Obstructive sleep apnea is a disorder characterized by repetitive
upper-airway obstruction leading to hypopneas/apneas during sleep. The diagnosis of hypersomnia
is based on chronic excessive daytime sleepiness for at least 3 months with concomitant
adequate/prolonged nighttime sleep duration. Restless leg syndrome is a disorder with the presence
of a severe urge to move legs accompanied by uncomfortable sensations with alleviation in leg
movement. Maintaining an appropriate amount and quality of sleep can lower the risk of developing
multiple disorders, as well as alleviate existing ones.
People with ADHD are at a higher risk of developing sleep disturbances than the general
population [57]. It may be since sleep disruption influences neurodevelopment in growing humans
as well as impairs neurocognitive functions, as mentioned before. According to research, children
with ADHD experience more wake after sleep onset time and longer latency of stage 3 of NREM sleep
than healthy counterparts [58]. In addition, the REM phase is shorter, and there is a smaller percentage
of time spent in the REM phase in comparison to the control group [59]. The most prevalent problems
among children include insomnia, night awakenings, and daytime sleepiness [60], whereas among
adolescents, insomnia, restless leg syndrome, and frequent snoring [61], and in the adult population,
night awakenings, daytime sleepiness, and psychosomatic symptoms during sleep onset [62]. Not only
does the disorder itself affect sleep quality, but also medications intended to treat ADHD, stimulants
like methylphenidate (norepinephrine-dopamine reuptake inhibitor), and non-stimulants like
atomoxetine (selective norepinephrine reuptake inhibitor) are shown to influence various aspects of
sleep. However, some ADHD medications were reported to improve sleep quality and decrease
nighttime awakenings with greater effects for methylphenidate [63]. They tend to change sleep patterns,
lowering the total REM and total NREM and increasing the sleep-onset latency (these results referred
to guanfacine (alpha-adrenergic agonist), methylphenidate, dasotraline (serotonin-norepinephrinedopamine reuptake inhibitor), L-theanine (glutamate reuptake inhibitor and a competitive low-affinity
glutamate receptor antagonist), and lisdexamfetamine (prodrug converting to dextroamphetamine –
norepinephrine-dopamine reuptake inhibitor – and L-lysine) [64]. At the same time, atomoxetine intake
did not cause any sleep pattern change [65]. Interestingly, atomoxetine has been successfully
administered in combination with oxybutynin to patients with obstructive sleep apnea. A combination
of noradrenergic and antimuscarinic signaling ameliorated the disorder's severity by improving the
responsiveness of the genioglossus muscle [66].
Not only does ADHD and its treatment directly affect sleep, but it also significantly contributes
to behaviors that are unfavorable to the quality of sleep. Patients with ADHD are more prone to
becoming addicted, as the disorder drives risky behaviors. Results of meta-analysis by Rohner et al. and
colleagues showed that more than twenty percent of substance use disorder patients manifest ADHD
symptoms. It was reported that people with ADHD, after drinking alcohol, had poorer sleep than their
peers without ADHD in the same setting. The same study examined the influence of cannabis, showing
no significant change depending on ADHD presence [67]. Another addiction more prevalent among
ADHD patients influencing sleep is smartphone and internet addiction, which causes greater severity
of insomnia, anxiety, and neuroticism in young adults [68]. For those who have internet gaming
disorder, ADHD seems to aggravate the eveningness of chronotype and insomnia [69], [70], negatively
affecting sleep quality. Also, food addiction, common among ADHD individuals, is connected with the
higher eveningness chronotype [71]. Additionally, ADHD is a risk factor for obesity[72]; a high-fat diet
predisposing to obesity induces REM sleep fragmentation by dysregulating the dopaminergic signaling
pathway. [73].
Importantly, there is a bidirectional relationship between ADHD symptoms and sleep
disturbances. Poor sleep quality, which can result from ADHD, in turn, worsens the ADHD symptoms,
forcing behavioral consequences, such as increased hyperactivity, impulsivity, inattention, cognitive
effects [74], such as increased hyperactivity, impulsivity, inattention, cognitive effects, like impaired
executive function, memory deficits, or reduced academic performance [75], and long-term health
implications including increased risk of comorbid conditions including depression or metabolic
disorders [76], [77]. Analogically, treating comorbid sleep problems can result in ameliorating ADHD
symptoms. Fadeuilhe et al. showed that the effect can be noticed after 6 months of treatment
introduction for comorbid insomnia disorder. Furthermore, the introduction of treatment for sleep
disorders resulted in better ADHD outcomes, reducing its severity [78].
The intricate interplay between sleep, ADHD, and its bidirectional consequences appears to be
inextricably linked and prone to change. The underlying mechanism involved in the connection has not
been widely examined. However, immunological pathways may provide some insight into the
workings of these interactions.
Comment 4:
In section 4, Mechanisms linking sleep and immunology: Lacks cohesion; paragraph flow is disjointed
with abrupt topic switches; Overlaps with previous section without clear delineation; Include a concise
summary table of immune changes under sleep sufficiency vs. deprivation.
Response 4:
Thank You for the notice. When the topic switches occur, their goal is to divide information based on
the studies cited to show which immune part has been connected with others. To make it easier to
follow we included subheadings. The recommended table is presented at the end of the paragraph.
4.1 Immunology of healthy sleep
The crosstalk between sleep and immunology occurs in health and disease. An adequate
amount of sleep, along with its quality, provides support for both the non-specific immune response
(cytokine production) and the specific immune response (T-cell function). Nighttime sleep, in
comparison to wakefulness, reduces the number of monocytes, NK cells, and all lymphocyte subsets.
At the same time, it increases the IL-2 production by T-cells but not IL-1beta, TNF-alpha, or IL-6 levels.
These effects are seen to be independent of cortisol changes [79]. Sleep reduces T-cell amount but also
affects the ratio – early nocturnal sleep imposes a change in T-cell helpers (Th) type with enhanced Th1
activity, which in later stages of sleep is replaced by Th2 dominance [80]. Similarly, the complement
system during sleep in healthy individuals is enhanced l [81]. Those and other changes that undergo
profound alterations in response to the circadian cycle help regulate the immune system, preparing for
a new day. After vaccination or during inflammation, sleep acts as a natural adjuvant protection and
contributes to the strengthening of the immune response [82]. Sleep quality is associated with
neurocognitive conditions via IL-18 and IL-12 [83]. Besides regulating cytokines, proper sleep affects
blood cell morphology. White blood cells and granulocytes were found to correlate negatively with
sleep efficiency. In addition, the granulocyte level correlated negatively with REM latency [84].
4.2 Influence of immune impairment on sleep
Excessive or prolonged immune activation due to impaired regulation or disease may
deteriorate sleep structure and efficiency [85]. Even the most common general symptoms, including
fatigue, fever, or pain, result in a decrease in short-term sleep quality and quantity. Prolonged
symptoms induce more severe changes, such as chronic non-restorative sleep with altered sleep
architecture and hypersomnia/insomnia, accompanied by excessive daytime sleepiness resulting from
this [86], [87], [88]. In addition, many autoimmune diseases, such as rheumatoid arthritis [89],
inflammatory bowel disease [90], multiple sclerosis [91], or psoriasis [92], seem to have a detrimental
effect on sleep. One-third of patients with autoimmune diabetes were reported to have altered sleep
quality, reflected as insufficient total sleep time [93].
4.3 Influence of sleep disturbances on the immune system
In contrast, sleep deprivation induces unfavorable changes in the immune system, leading to
increased susceptibility to infections and a chronic inflammatory state. Partial sleep deprivation results
in temporarily reduced mitogen-induced cell proliferation, lower levels of HLA-DR expression,
increased CD14 expression, and changes in CD4 and CD8 levels [94]. There is also a significant decrease
in IL-6 levels [95]. Acute loss of sleep predisposes to cardiovascular diseases as it elevates the level of
catecholamines [96]. Chronic sleep restriction, such as working night shifts, also exacerbates these
changes [97]. What is important, the chances seem to be reversible after recovery sleep or napping [98],
but not entirely, as Th cells and IgA levels were reported not to be reestablished after sleep recovery
[99]. Sleep deprivation decreases the melatonin level, which may contribute to further disruption of the
circadian rhythm [100]. Moreover, it participates in neurodegenerative disease development via
oxidative stress exacerbation [101]. Sleep insufficiency is also a risk factor for gut microbiota dysbiosis
[102]. Furthermore, sleep deprivation was reported to induce an elevated granulocyte count with no
statistically significant changes in other parameters [84]. As presented, sleep generally acts as one of the
primary regulators of the immune system, and its quality influences the direction of immune
development. Not only sleep but the entire circadian rhythm also affects human immunology, with
interactions present in both directions. Thus, seeing that ADHD is connected with sleep disorders and
immunology on multiple levels, and immunology is profoundly directed by sleep, one can look for
some dependencies between ADHD and both sleep and immunology.
Table 2 – immune change in healthy sleep vs sleep insufficiency
Immune change in healthy sleep vs sleep insufficiency
Category Healthy sleep Sleep insufficiency
Cytokines
Affect within physiological limits
with a marked increase in the IL-2,
IL-12, and IL-18
A decrease in the IL-6
Cells
Affect within physiological limits
with a marked decrease in the
granulocytes, NK cells, and all
lymphocyte subsets, with first Th1
and then Th2 dominance
An increase in
granulocytes
MHC Affect within physiological limits Lower HLA-DR
expression
Gut microbiota Affect within physiological limits Dysbiosis in longterm
Melatonin Affect within physiological limits A decrease in the
melatonin levels
Stress hormones Affect within physiological limits
An increase in the
level of
catecholamines
Abbreviations: IL – interleukin, MHC – major histocompatibility complex, NK –
natural killer, Th – T-helper
Comment 5:
“In section 5, Relationship between sleep and immunology in ADHD: Provide a clear conceptual model
illustrating how circadian gene dysfunction, immune activation, and ADHD symptoms intersect
(beyond Figure 1, which is too general); Reduce the exhaustive listing of genes and SNPs.
Response 5:
Thank You for the comment. We improved Figure 1, which has more details as You recommended by
adding potential influences on each other. For the genes and SNPs list, the aim was to collect all
available data on the topic, which includes associated information on genes and SNPs, thus omitting
them would decrease the informative value of the manuscript.
Figure 1 – The associations between sleep, ADHD, and immunology
Abbreviations: ADHD – attention deficit hyperactivity disorder; HPA – hypothalamicpituitary axis; IL – interleukin; REM – rapid eye movement; TNF – tumor necrosis factor.
Comment 6:
In section 6, Therapeutic Interventions: Discuss limitations of current treatments and highlight gaps
in evidence linking immune-modulatory therapies to ADHD; Suggest testable hypotheses or future
clinical trial designs targeting sleep-immune-ADHD nexus.
Response 6:
Thank You for the advice. At the end of the paragraph, we added limitations, gaps, and hypotheses
mentioned above.
However, the mentioned therapeutic interventions have their cons, including delayed
response to treatment, especially in non-pharmacological treatment – CBT and sleep
hygiene/lifestyle change, with reduced general attention span, inconsistent routine, and executive
dysfunction typical in ADHD subjects. Side effects, such as mood changes and risk of dependence,
are common in pharmacological treatment. In addition, they may interact with other medications
used. Finally, while using any option – pharmacological and non-pharmacological – one should not
overlook other comorbidities, as there can be a variability of effectiveness. Having in mind those
limitations may contribute to a more personalized approach and a better response to the
implemented therapy. All of the treatment options comprise more sleep-focused interventions than
immune ones, as there is a gap in evidence linking directly to immune-modulatory therapies and
their effect on sleep and ADHD symptoms. Currently inconclusive, but in need of more research,
hypotheses include the influence of classic anti-inflammatory agents on sleep and the ADHD
condition, and the influence of probiotic supplementation on both sleep and ADHD.
The treatment plan for each ADHD case should be considered individually, tailored to the needs
of each individual, in accordance with the latest recommendations. The best results are mainly achieved
by combining properly chosen ADHD medication and non-pharmacologic interventions. Every change
in the treatment plan should be clearly explained and discussed with the patient, as it may be a challenge
to fulfill the recommendations. The summary of this information is depicted in Figure 2.
Comment 7:
While generally informative, the manuscript contains multiple grammatical inconsistencies, awkward
phrasing, and typographical errors. A thorough professional language edit is strongly recommended.
Response 7:
Thank you for the comment. The manuscript was edited by a native speaker.

Round 2
Reviewer 2 Report
Comments and Suggestions for Authors
The authors have improved the quality of their review in this revised manuscript. The authors have addressed all my comments and critiques in detail, thereby I recommend this manuscript for publication.